# Using Neural Networks to Estimate Site-Specific Crop Evapotranspiration with Low-Cost Sensors

**Jason Kelley [1],\*** **and Eric R. Pardyjak [2]**

[1] Department of Soil and Water Systems, University of Idaho, Moscow, ID 83844, USA
[2] Department of Mechanical Engineering, University of Utah, Salt Lake City, UT 84112, USA; pardyjak@mech.utah.edu
\* Correspondence: jasonrk@uidaho.edu; Tel.: +1-208-885-1015

**Abstract:** Irrigation efficiency is facilitated by matching irrigation rates to crop water demand based on estimates of actual evapotranspiration (ET). In production settings, monitoring of water demand is typically accomplished by measuring reference ET rather than actual ET, which is then adjusted approximately using simplified crop coefficients based on calendars of crop maturation. Methods to determine actual ET are usually limited to use in research experiments for reasons of cost, labor and requisite user skill. To pair monitoring and research methods, we co-located eddy covariance sensors with on-farm weather stations over two different irrigated crops (vegetable beans and hazelnuts). Neural networks were used to train a neural network and utilize on-farm weather sensors to report actual ET as measured by the eddy covariance method. This approach was able to robustly estimate ET from as few as four sensor parameters (temperature, solar radiation, humidity and wind speed) with training time as brief as one week. An important limitation found with this machine learning method is that the trained network is only valid under environmental and crop conditions similar to the training period. The small number of required sensors and short training times demonstrate that this approach can estimate site-specific and crop specific ET. With additional field validation, this approach may offer a new method to monitor actual crop water demand for irrigation management.

**Keywords:** machine learning; site-specific; actual evapotranspiration; irrigation efficiency

## 1. Introduction

Efficient irrigation depends on reliable crop water demand information to prevent drought stress and maximize yield from available water resources. One direct measure of crop water consumption is actual evapotranspiration ($ET_a$). Precision irrigation technology depends on localized, spatially explicit and real-time estimates of crop water needs to achieve the potential efficiency gains offered by precision methods. Growers and irrigation specialists currently have many information resources including historical water records, regional weather networks and remotely-sensed estimates of weather and water demand [1]. Irrigation research and weather monitoring networks facilitate improved irrigation by providing detailed information about actual crop water demand [2]. However, appropriately localized and real-time estimates of crop water demand are either unaffordable or unavailable [3]. Because the risk of yield reduction or crop failure are highly dependent on timing of water application, real-time measurements of water demand are necessary [4,5]. Further progress in mapping crop water demand will depend on new measurement technology and methods. For these measurements to be useful to producers, they must be affordable, accessible and compatible with existing irrigation systems and farming practices [6]. Site-specific, real-time ET data provides precise information to support irrigation scheduling, facilitating optimal allocation decisions, mitigating crop loss risks and improving overall irrigation efficiency.

Irrigation allows arid regions to be among the most agriculturally productive per unit area and material inputs, yet uncertainty about water supply adds an additional dimension of risk [7]. Climate change, drought and water allocation rights are all important drivers of water availability and must factor into crop and irrigation planning [8]. Likewise, advances in precision irrigation can potentially reduce water use while maintaining yields but these systems are similarly constrained by an absence of local real time data at the field scale [3]. In practice, implementation of science based irrigation strategies, such as deficit irrigation; site specific or variable rate irrigation requires reliable data about crop water demand. In the case of incremental water use reductions, the risks of under-watering can outweigh any cost of over-watering [4]. This is especially true during periods when ET estimates are more uncertain, such as with deficit irrigation or extreme weather conditions (drought and heat), when atmospheric, soil or crop behaviors do not conform to critical method assumptions (such as a adequately-watered surface). Relying on regional weather network data, science-based estimates of crop water demand are often not localized enough to make incremental water decisions within the framework of a farm's overall water planning [9]. Every farm venture must manage inherent risk from weather and crop variability. Irrigators make daily and hourly decisions based on many sources of information including agricultural weather networks and on-farm weather stations.

Extensive work at the state and regional level has been put into developing publicly accessible irrigation information [10–12]. Most notably, state and regional weather networks include automated calculation of reference evapotranspiration ($ET_{ref}$) using the Penman-Monteith equation. Reference ET can be used with calibrated crop coefficients to estimate a crop specific ET. In the Pacific Northwest, the US Bureau of Reclamation operates the Agrimet network with approximately 70 agricultural weather stations transmitting hourly weather and $ET_{ref}$ data (www.usbr.gov/pn/agrimet). California's Dept. of Water Resources operates the CIMIS network of 145 weather stations, including spatial mapping of 18 statewide zones for $ET_{ref}$ (www.cimis.water.ca.gov). Washington State operates AgWeatherNet with 172 automated stations (weather.wsu.edu). While these networks are a valuable resource, they do not account for all spatial variability in crop development, microclimate or water availability. Neither do they provide a simple way to account for effects of site-specific conditions such as soil water characteristics, non-optimal irrigation, novel water management strategies or new crop varieties for which crop coefficients have not yet been calibrated [13].

When growers use scientific irrigation planning to maximize water use efficiency, the benefit of reduced water consumption can be linked to greater risk of reduced yield or crop failure. Under drought stress, plants can limit water loss so that actual transpiration is less than potential evapotranspiration. Once a crop is planted, farmers often must prioritize water allocation, deciding which crops should receive full irrigation and when to under-irrigate. Yet these decisions are usually made without reliable information on actual ET or crop water demand, particularly at scales that are relevant to precision irrigation systems [14]. These decisions must also be coordinated with distribution capacity and source water availability. Coordinating all of these factors in real time is an extraordinary challenge. At present, farmers who use evapotranspiration as a basis for irrigation scheduling rely on daily reference ET estimates and single crop coefficients provided with network data. Consequently, farmers must augment regional ET estimates with their personal knowledge of local weather patterns, crop water needs, microclimate and soil water storage to help plan irrigation decisions.

Regional agricultural weather networks are a valuable source of information but even the densest networks leave large gaps in coverage. Local on-farm conditions may vary significantly from those at the monitoring site, leading some growers to construct their own on-farm weather stations. Commercially available systems do provide site-specific information on weather parameters but do not necessarily have the quality of sensors of the typical network station. Some commercial systems can be accessed with smart phone interfaces and may calculate reference ET. Another challenge for using low-cost, on-farm stations is requirement for a homogeneous, well-watered surface, which ensures boundary layer conditions that are representative of reference ET [13,15]. Stations are not usually constructed with a reference surface because the cost of maintaining such a site can cost more

than the station itself. In fact, most farm weather stations are located next to buildings or along farm roads so that they do not interfere farm operations. Maintaining a reference surface and siting the station in an appropriate wind fetch are two critical considerations that are provided at permanent network sites but difficult to reproduce under normal farm operating conditions.

Once on-farm weather data (temperature, humidity, solar radiation, precipitation, etc.) is collected, a set of procedures must be selected for quality control and assessment (QAQC) and to estimate crop water demand. If on-farm stations do not provide reliable and useful information, the value of the stations do not justify the cost of maintaining them. Even for cases in which advanced control systems are used, the impact of poorly calibrated crop coefficients and in-homogeneous soil characteristics confound efficient irrigation scheduling [16]. Regardless of the source, when operators attempt but fail to find direct uses for weather data in farm operations, confidence in scientific methods for managing water is eroded and can severely impair collaborative efforts to manage water resources [8]. Moreover, when on-farm weather sensors are used to determine reference ET, crop coefficients are still needed to determine actual ET and prescribe an irrigation application rate [17]. Direct measurements of actual ET are possible through more laborious and costly research tools such as eddy covariance or weighing lysimeters, yet these methods are usually infeasible for monitoring water consumption in applied settings. In cases when this information is most valuable and when resources are truly scarce, such as when improving agriculture systems in rural areas, information obtained by such methods is difficult to obtain or apply directly [18].

In order to leverage all of these techniques, neural networks show potential to integrate different kinds of ET measurement methods. Artificial Neural Networks (ANNs) are computational models that allow computers to learn from data and approximate solutions for non-linear, multi-input functions. Due to their flexibility and robustness, ANNs have been widely applied in fields ranging from robotics to airplane flight control [19,20]. ANNs have seen exploratory usage in hydrology [21] and specifically in modelling reference ET [22] and irrigation optimization [23]. Because neural networks use optimization algorithms and do not depend on physical models, ANNs can be useful in circumstances where data quality is variable and also to explore non-linear behaviors. Data from a previous field study on crop water use was used to train an ANN to estimate ET and examine the robustness of the method in both well-watered and drought conditions, complementing and extending the Penman-Monteith equation for reference ET [24].

The two central objectives of this effort are to augment the value of on-farm weather data by making them more directly useful in irrigation scheduling and to facilitate more extensive ET monitoring. By co-locating eddy covariance sensors with more readily available meteorological sensors, neural networks can be employed to estimate actual ET from the same parameters typically used to determine reference ET. The project is used to demonstrate that an affordable, on-farm weather station can determine site-specific $ET_a$ when co-located with research grade sensors. Short periods of field data from co-located eddy covariance and weather stations were used to first train the neural network to estimate $ET_a$. The resulting ANN estimate of $ET_a$, based on low-cost sensor data, was compared against the eddy-covariance data for robustness and accuracy for the entire field record. This approach was used to (1) evaluate the minimal sensor requirement for robust ANN; (2) determine the shortest reliable training period; (3) provide new $ET_a$ observations that can guide more efficient irrigation. By training for short periods with actual ET data, neural networks and simple sensors can be used to estimate actual ET. By sharing higher-cost research sensors across multiple sites, actual ET can be measured in many more locations than would be possible with either eddy-covariance or standard agricultural weather networks alone.

## 2. Materials and Methods

Development and validation of the neural network method to estimate actual ET was performed in two phases. First, field studies were conducted at a center-pivot irrigated field and at a drip irrigated orchard using co-located standard weather stations and eddy covariance systems. Post-season analysis

consisted of automated training of neural networks and subsequent determination of the shortest viable training periods and the minimal number of required sensors. These determinations were made by dropping out data and comparing the resulting ET estimate to the entire collected record for that site. The resulting networks were then used to estimate actual ET and to investigate irrigation performance and crop response at each site. The trained ET estimates were evaluated for robustness under crop water stress and in environmental conditions typically encountered under normal farming conditions. These included non-optimal irrigation scheduling, periods of heat stress, long periods of reduced solar radiation from wildfire smoke and irrigation of crop with unknown irrigation demand (crop coefficients). In each case, the ANN estimate of actual ET ($ET_{ann}$) was compared to actual ET determined by eddy covariance. Where applicable, $ET_{ann}$ was also compared against crop ET determined with crop coefficients and $ET_{ref}$.

Data used in this study and Matlab code implementation of the neural network analysis are maintained by the Northwest Knowledge Network in a publicly accessible database. The data can be obtained with the following DOIs: www.doi.org/10.7923/7nt0-7e64 [25] (site 1 data) and www.doi.org/10.7923/nnxk-8p22 [26] (site 2 data).

## 2.1. Field Experiments

The two experimental sites are located in irrigated fields located in western Oregon, USA. The first field site (site 1) is located in Benton County Oregon, USA, between two fields irrigated by center pivots. The eddy-covariance system and weather station were sited between the two fields, adjacent areas of which were planted in green beans (*Phaseolus vulgaris*) covering 22 ha and 29 ha in the two fields. Eleven years of wind records were available from a municipal airport weather station 2 km away from the measurement site. From these data, the predominant summer wind patterns were the basis of siting and orienting the weather sensors. To maximize the likelihood of reliable eddy-covariance measurements, the measurement footprint is uniformly flat, irrigated and planted to crops for at least 200 m (100 × the measurement height) in the direction of the predominant day time wind direction and nearly as uniform for 200 m in the directions of all other wind (night time katabatic flows and most common synoptic scale weather patterns), Despite the center pivot irrigation operating for the previous four years, for the study in 2017 a consistent irrigation schedule had not yet been established by the farm manager. Instead, irrigation timing was based on visible crop stress and hand tests of soil moisture; application rates were limited by soil intake rates (personal correspondence).

Data were collected for eight months (March–October 2017) with a variety of sensor configurations. All data used in this analysis were collected for the entire eight-month period. For eddy-covariance measurements, an IRGASON integrated sonic anemometer and open path gas analyzer (Campbell Scientific Inc., Logan, UT, USA) was mounted at 2 m above ground level (a.g.l.) and oriented into the dominant daytime wind direction. For some 2–3 week training periods, a NR01 net radiometer (Hukseflux, Delft, The Netherlands), HFP-01 soil heat flux plate (Hukseflux, Delft, The Netherlands) and two HCS2 humidity/temperature probes (Rotronic AG, Bassersdorf, Switzerland) were co-located with the Q-7 net radiometer soil probes and HMP-60 thermohydrometers, respectively. The "low-cost" sensor ensemble included: two humidity and temperature sensors at two measurement heights; soil moisture content temperature potential sensors; tipping bucket rain gauges; sonic anemometers; a Q-7 thermopile net radiometer and a PAR quantum sensor; a cup and vane anemometer. Specific sensors at the two sites are described in Table 1. A typical sensor array used in the field experiments is shown in Figure 1.

A similar combined sensor system (weather sensors and eddy covariance) was deployed at site 2 in summer 2018, which was located in Yamhill County, Oregon USA, at a 38 hectare (94 acre) drip irrigated hazelnut (*Corylus avellana*) orchard. The orchard was planted in 2008, had been irrigated since planting and had reached full production in 2017. The orchard manager reported that he had applied increasing amounts of water each year without a clear endpoint for full irrigation and in 2018 was applying 20 mm (3/4 in) weekly (over three weekly applications). Irrigation is also used to apply fertilizer

(fertigation). A field experiment in 2017 (exclusive of eddy covariance) provided preliminary weather information such as predominant wind direction, which was used in siting this field experimental. In the summer of 2018, Site 2 was instrumented with the IRGASON eddy-covariance system and supplementary sensors. Sensors were mounted on a mast/boom system at approximately twice the canopy height (Figure 2).

**Table 1.** Description of low-cost sensors used in the on-farm experiments.

| Measured Parameters | Sensor (Manufacturer) | Site 1 | Site 2 |
|---|---|---|---|
| Air Temperature and Humidity | HMP60 (Vaisala, Vanta, Finland) | 2 m a.g.l. and 5 m a.g.l. | Canopy Top & 3 m above canopy |
| Soil Moisture & Temperature | GS3 (Decagon Inc., Pullman, WA, USA) | 5 cm and 30 cm depths | At drip emitter, & 25 cm from emitter (5 cm depth) |
| Soil Matric Potential and Temperature | MPS-2 (Decagon Inc., Pullman, WA, USA) | 5 cm depth | 5 cm depth, near drip emitter |
| Precipitation and Irrigation | ECRN-100 (Decagon Inc., Pullman, WA, USA) | Gauges within & outside irrigated area | Captured drip irrigation |
| Net Radiation | Q-7 net radiometer (REBS, no longer in production) | 2 m a.g.l. | Top of canopy |
| Downwelling Photosynthetic Active Radiation | PAR quantum sensor (Apogee Instr., Logan, UT, USA) | 2 m a.g.l. | N/A |
| Wind Speed and Direction | DS-2 Sonic anemometer (Decagon Inc., Pullman, WA, USA) | 2 m a.g.l. | N/A - used IRGASON anemometer data |
| Wind Speed and Direction | Wind Sentry cup and vane (RM Young, Traverse City, MI, USA) | 2 m a.g.l. | N/A |

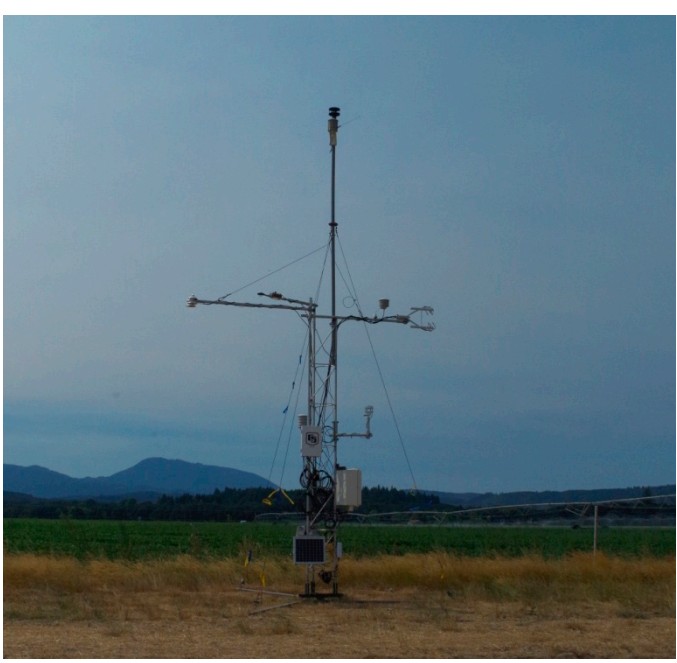

**Figure 1.** Typical sensor array used in the field experiments shown at site 1. Crop (green beans) and center pivot irrigation system is visible in the background. Note that this is not the full array used for the 8-month period at site 1 but an auxiliary system that is shown for clarity. The eddy-covariance system used here (IRGASON) is visible extending to the right.

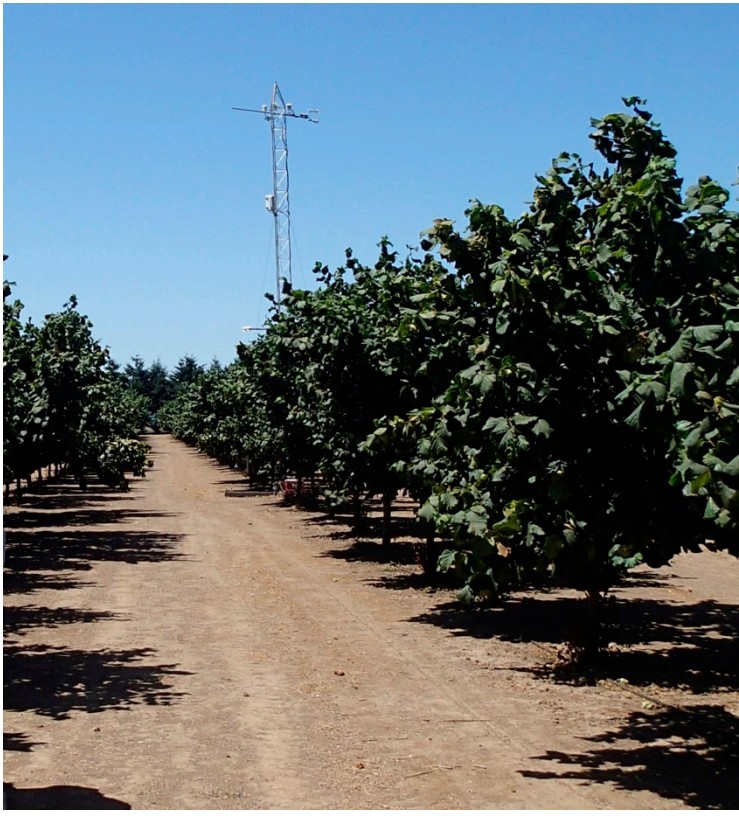

**Figure 2.** Photograph of experimental site 2 (Hazelnut orchard) showing the eddy-covariance system, net radiometer and other sensors on a 10-m tower in background.

Raw sensor outputs were measured and logged with CR1000 dataloggers (Campbell Scientific). IRGASON data were logged at 20 Hz and all other sensors were sampled on 1 s intervals, with 1-min averages recorded. All Decagon sensors were sampled and recorded with a Decagon EM50 datalogger at the fastest averaging interval of 5 min. Eddy covariance provides the control estimate of $ET_a$ used for ANN training. For eddy-covariance measurements, wind speed, sonic temperature and water vapor density are sampled at 20 Hz. To ensure representative ET measurements, standard eddy-covariance corrections and data quality procedures [27], 2-d coordinate rotations [28] and density corrections [29] were employed. A phase space method of despiking was employed [30]. Eddy-covariance measurements of ET were calculated for 15 to 60 min averaging periods. The net radiation, soil heat fluxes, sensible and latent heat fluxes were used to evaluate closure of the surface energy balance as a secondary estimation of uncertainty in the ET control measure. Periods which exceeded 20% residual were excluded from ANN training and in analysis. Limited gap filling was required for short periods (~1 averaging period), which were linearly interpolated for calculating cumulative ET.

The local reference ET was also calculated for the analysis, using the Penman-Monteith equation [31] based on the measured net radiation, ground flux, air pressure, temperature humidity and wind speed. $ET_{ref}$ was used as an approximation of potential ET for identifying time variability in irrigation adequacy and to identify periods demonstrating non-linear crop responses, that is, when ET was limited by water availability rather than available energy. $ET_{ref}$ was also used in combination with single crop coefficients to compare to $ET_{ann}$, as the single crop coefficient method is a readily available ET estimate used for irrigation scheduling. Where applicable, crop coefficients were obtained from Agrimet [32] (https://www.usbr.gov/pn/agrimet/cropcurves/crop_curves.html). Energy balance closure was used as an independent check on data quality to ensure that eddy-covariance measurements were representative of actual field conditions. Fluxes were compared against irrigation

timing, soil moisture measurements and atmospheric stability. For both sites, soil moisture and precipitation gauges were installed so that irrigation and precipitation were measured independently (using multiple tipping buckets), so that the total precipitation and irrigation intercepted by the crop were recorded.

## 2.2. Estimating Actual ET with Artificial Neural Networks

Once a time series of actual ET was established using quality controlled eddy-covariance measurements, raw sensor data and ET control measurements were imported and processed in Matlab [33]. The neural network used to estimate actual ET was implemented with MathWorks' Neural Network Toolbox. A simple two-layer network (with one layer of 10–12 nodes and one single node output layer) was trained using the Bayesian Regularization backpropagation algorithm, which is implemented with Levenberg-Marquardt optimization. This algorithm is well suited to moderately sized data as is typical for the experimental design described above. Additionally, the Bayesian regularization algorithm is robust when using noisy input data [34].

The structure of an artificial neural network is built of nodes (also called "neurons"); each node is an array of input parameters and weighting and bias vectors control nodal input and outputs [21]. In the training process, the network is adjusted by changing the functional contribution of each input parameter through assigning unique weighting and biases at each node. The network uses feedback from the control estimate (in this case, ET measured by eddy covariance) to train functional relationships. From random initial conditions, training proceeds by iterating and minimizing the difference between the control ET measurement and ANN estimated values. Training ceases when a specific error level is reached, a specified number of iterations is completed or when error decreases negligibly in successive iterations. In the Matlab toolbox, the user specifies the data input, training data and algorithms for optimizing the ANN; processing occurs in an intuitive user interface which also allows for plotting results and evaluating the robustness of the resulting solution. Every iteration of the ANN is randomly initialized, so that two ANNs trained on identical data will produce slightly different outputs.

The field experiments were designed to evaluate the feasibility of using ANNs working from the hypothesis that the most relevant information would be obtained from sensor data related to physical principles such as conservation of energy (surface energy budget) and turbulent flux relationships. Accordingly, two practical aims were proposed. The first aim was to determine which low-cost sensors are most critical for robust ET estimates and cost effective for monitoring crop water demand. The second aim was to determine the shortest training time that is required for a valid ANN solution. The motivation of these two aims are to reduce direct costs and to extend the impact of eddy covariance sensors (i.e. to train as many independent weather stations as possible).

Sensor selection was based on a hypothesis that parameters related to the largest terms in the surface energy budget (net radiation, sensible heat flux and soil heat flux) would be the most important parameters for robust ANN solutions. Accordingly, the field experiments included sensors for the main common meteorological parameters: air and soil temperature, incoming and net radiation, air humidity and wind speed; wind speed also may provide information about the impact of advection. Likewise, additional temperature, humidity and wind speed sensors were placed at multiple measurement heights, corresponding to the idealized flux-gradient relationships. In addition to physics based approaches, non-linear drivers of ET such as water availability were considered. A secondary hypothesis was that time series of soil moisture, soil matric potential, precipitation and irrigation and wind direction would help increase the skill of the trained neural network to resolve site specific ET patterns.

The second aim was to evaluate the required time to train robust ANN models. Prior to implementation, it was hypothesized that two weeks of training data would be sufficient to reproduce observed $ET_a$ from raw sensor data. Training periods of 1–21 days were used on all field data and the modelled $ET_{ann}$ results were compared to the half hour $ET_a$ measured by eddy covariance during all half hour periods. The $R^2$ and RMSE statistics and the slope of the regression between $ET_a$ and

$ET_{ann}$ were compared to the same statistics comparing $ET_a$ and $ET_{ref}$. The slope of the regression line is equivalent to a single crop coefficient and it was expected to approach unity for the ANN method (a trivial result). A successful training time was noted based on $R^2$ and root mean square error (RMSE) comparable to those found for $ET_{ref}$. In other words, a successful training period needed to produce a time series at least as robust as that found with the Penman-Monteith equation, using eddy covariance as the control measurement of actual ET. Also, half-hour periods in which the surface energy budget residual exceeded 20% of the total available energy were excluded from training set (but included in error analysis).

This project used the simplest ANN implementation in the Matlab Machine Learning Toolbox (MathWorks, 2016). The ANNs employed in this study used a single layer of 10 "hidden nodes" and one node in the output layer. Training was conducted using the Levenberg-Marquardt training algorithm, with weights and biases regularized assuming a Bayesian distribution of these parameters. Other configurations were tried, including larger and smaller numbers of nodes, layers and different training algorithms. Typically, the simplest ANN structures performed the most reliably and a general strategy to prevent overfitting of the ANN model relies on limiting input parameters [35,36]. This may be due to the relatively small number of input variables (relative to typical ANN applications).

The workflow for training the neural network and estimating $ET_{ann}$ was as follows:

1.  Half-hour mean values of each input parameter were calculated. Time of day was determined from the data record time stamp. In later iterations, the timing of specific irrigation events was used and was calculated manually as a separate time series based on field notes and irrigation records and corroborated by precipitation gauge and soil moisture data.
2.  The training period was set based on a starting date and a number of days.
3.  The network nodes and layers were specified. Typically, this was set and an arbitrary but small number of 10 nodes in a single hidden layer. The network also assigned a single node output layer.
4.  Training proceeded automatically using the Matlab toolbox, with 90% of data rows in the training set (each half hour record is a row) used for training and 10% used for testing. Training consists of a random assignment of initial weights and biases in the hidden nodes, which were adjusted iteratively until the MSE between the training set and $ET_a$ (from eddy covariance) no longer improves (or until a maximum number of iterations was reached).
5.  Half-hour averages for the entire data record were then input to the trained ANN model, which estimates the time series of $ET_{ann}$.
6.  Training was repeated for each network arrangement (number of nodes, training time, input parameters) to evaluate hypotheses about the repeatability and robustness of ANN ET estimates.

### 2.3. Using the ANN Method to Monitor Crop Water Use: Two Case Studies

While operational estimates of daily reference ET are widely available, the difference between actual and the locally calculated reference ET is inextricably linked to potential water savings and the potential to incur crop water stress [1]. The departure of actual from reference ET is a non-linear function and affected by soil properties, crop drought adaptability and by water availability. Estimates of this departure ($ET_{ref}$-$ET_a$) on a daily or hourly basis provides an objective metric of irrigation adequacy and can identify the site-specific and variety-specific crop water requirement, optimize scheduling and evaluate the effectiveness of irrigation strategies such as heat mitigation and deficit irrigation. Supporting objectives for this study were to determine the minimum required length of training (by reducing the amount of data provided for training) and to determine which combination of sensors is cost-effective and robust at estimating ET when trained (by eliminating sensors from input arrays).

## 3. Results

Neural networks were successfully used to estimate actual ET at both study sites. Results obtained from two different crops show that ANNs can be trained on eddy-covariance data to determine $ET_a$

from low-cost weather sensors. The resulting daily and cumulative water use estimates are at least as accurate than the Penman-Monteith (P-M) equation for well-watered crops (with known crop coefficients). Under drought stress conditions or where there are no established crop coefficients, the ANN could reliably predict $ET_a$. In each case, a minimal suite of low-cost sensors was optimized and new simple parameters were identified which bolster the ANN training without incurring additional cost or complication. Minimum training times of three days to two weeks were found in each case, although other considerations of training were shown to influence the validity of the resulting networks. At each location, site specific considerations demonstrated the utility of the ANN method to provide insights into variability in crop water use, information which can be used directly in irrigation decisions.

*3.1. Optimization of ANN Training*

Due to the numerical (rather than physical) basis of the neural network training, determination of training success was based on statistical comparison with the control measurement from eddy covariance. Latent heat flux estimated by the ANN is compared to the observed latent heat flux (from eddy covariance) for the entire time series. An excerpt of an output time series shows the close correspondence between modelled ANN and eddy-covariance measurements following the training period (indicated by the shaded times, Figure 3). Following the six steps outlined in the methods section, repeat iterations (step 6) of the ANN training produce similar results for a given training period. The resulting ET estimates were relatively insensitive to the number of sensor parameters used to train the ANN.

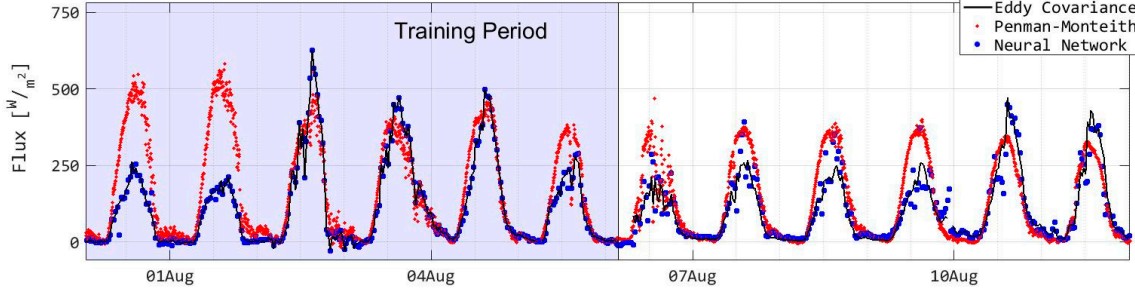

**Figure 3.** Excerpted time series of latent heat flux determined by the eddy-covariance measurements, Penman-Monteith equation (with no crop coefficient applied) and the actual latent heat flux estimated by the artificial neural network (ANN) method. The shaded area indicates the period used for training the ANN.

Typically, training began with a set of 10–12 input parameters including time of day, air temperature and humidity at two heights, wind speed and direction, net incoming solar radiation, soil temperature and heat flux, soil moisture and precipitation. Antecedent (lagged) conditions were also used. However, the most robust and computationally simple ANN trainings only required 4–6 input parameters, most often air temperature and humidity (at one height), net radiation (from the relatively low quality Q-7 radiometer), time of day and wind speed. It was initially hypothesized that inclusion of the soil moisture would allow the ANN to approximate non-linear behaviors related to water stress. Surprising, soil moisture and matric potential always reduced the stability and accuracy of the resulting ANN model (see discussion below), despite an apparent relationship between the time series of observed soil moisture, potential ET and actual ET. One possible explanation for this failure is that the time scale associated with changes in soil moisture are decoupled from the time scales of atmospheric transport and biological responses. If so, flux behaviors were not recognized by the ANN model used here, which evaluates each record (of all observed parameters) independently in time from other observations. An ANN model incorporating time series lags or decay functions bear further investigation.

Training results were also relatively insensitive to training times between 7–14 days for data from site 1. At site 2, training on fewer than 10 days resulted in poor model stability. Poor model results ($R^2$ and RMSE when compared to $ET_a$) often corresponded to a training that resolved in either very few iterations (resolved to a numerically stable but incorrect solution) or did not resolve before the maximum number of iterations was reached. The number of iterations ("epochs") was used as a qualitative indicator of ANN results. While a required training time of seven days was shorter than an initial guess of two weeks, training for shorter periods became unstable and the resulting 30-min estimates of $ET_{ann}$ at best showed no improvement over an un-calibrated Penman-Monteith estimate (Figure 4). It is possible that the simple structure used in this project made the resulting network more likely to revert to stable but incorrect solutions: as the training parameters approached lower limits (i.e., 3–5 days), the network became increasingly likely to stop training nodes on a stable result that predicted dramatically incorrect ET values through a process known as overfitting [21].

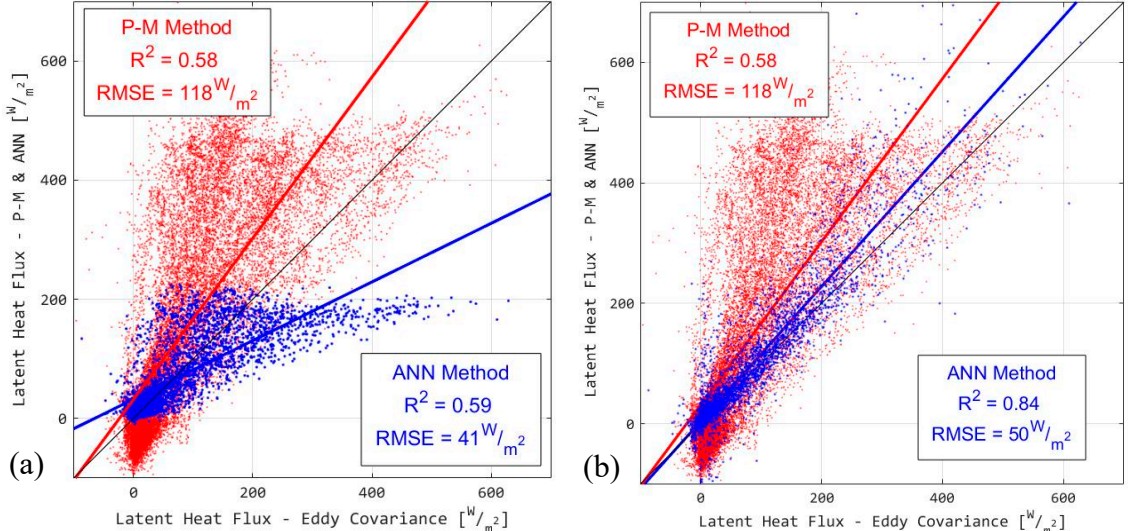

**Figure 4.** Results at site 1 from (**a**) 3-day training and (**b**) 14 day training period. The RMSE indicates the departure of the dependent values (P-M and ANN) from the eddy covariance control.

While an optimal training time was relatively easy to determine from multiple iterations, a more challenging consideration is that the resulting model accuracy was primarily influenced by the environmental conditions which occurred during the training period. The ANN estimates for any 7-day training period may be numerically robust but when compared to the entire time series, the ANN was only able to accurately predict flux under conditions which were similar to those observed during training periods. For example, if the ANN is trained during a period of adequate (or over) irrigation, then the resulting $ET_{ann}$ estimates will resemble full potential ET, regardless of actual conditions. Likewise, if the training data is only sampled from periods during which the crop exhibits drought stress, the resulting ET estimate will be accordingly too low (Figures 5 and 6).

Based on these results, the trained ANN models were used to monitor $ET_a$ and evaluate crop condition relevant to irrigation scheduling for the two sites. In each case, site-specific conditions demonstrated the value of real-time monitoring of actual ET. At site 1, the departure of $ET_a$ from $ET_{ref}$ corresponded to non-optimal irrigation so that the crop water use did not conform to standard models. At site 2, because there is no established crop coefficient for hazelnuts, the ANN provided a unique way to monitor actual ET.

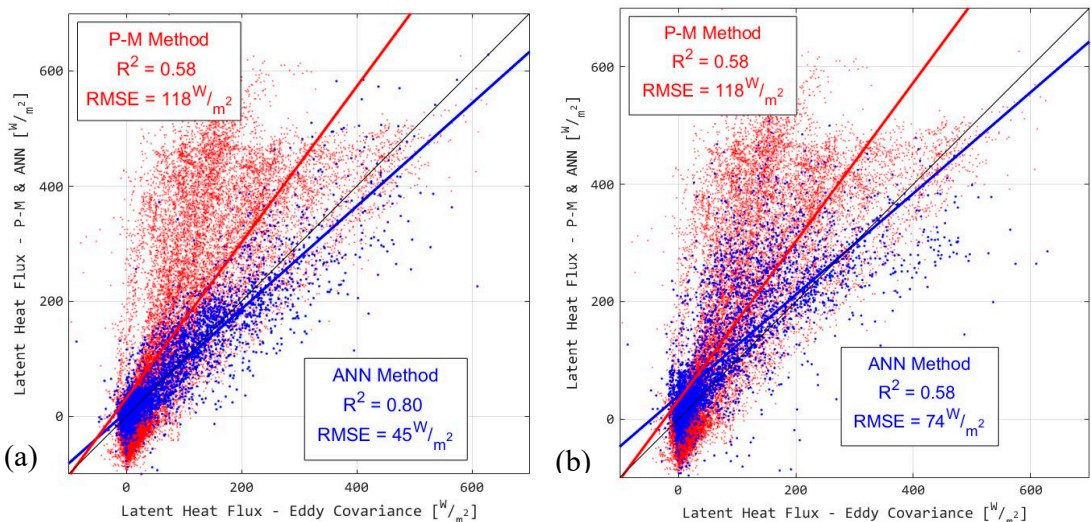

**Figure 5.** Method comparisons at site 1 from 7 day training (**a**) mid-season, when the crop was adequately irrigated; and (**b**) at the end of season, when irrigation was over-applied.

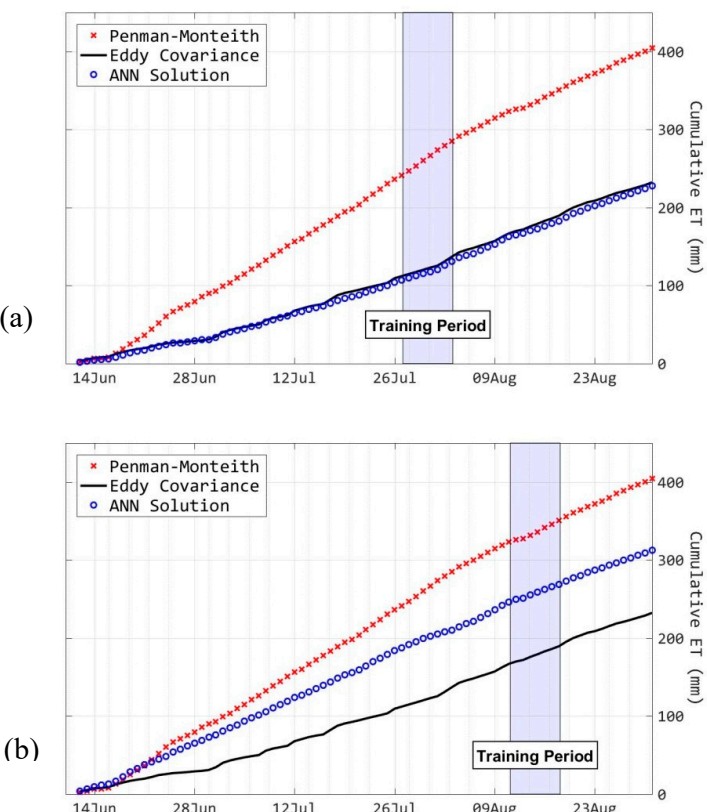

**Figure 6.** Training results at site 1 using 7-day training periods—Trained during (**a**) midseason and (**b**) end of season. Although half hour comparisons for the two trainings do not appear significantly different (Figure 5), the cumulative water use plot shows that training during a period of over watering (**b**) results in the ANN estimating significantly greater ET over the entire two month growing season.

### 3.2. Effect of Irregular Irrigation on Actual ET at Site 1

Direct measurement of actual $ET_a$ can be used to evaluate the effect of water availability on consumptive use. In the case of site 1, irrigation was adjusted throughout the season in an attempt to compensate for apparent water demand (based on the irrigators' hand tests of soil moisture, crop

development and in anticipation of weather forecasts). However, the irrigator expressed that lack of information on the actual water demand meant that these adjustments were largely subjective and led to inconsistent interannual yields and even crop failure in some years (personal communication). In the 2017 growing season, the field at site 1 was irrigated weekly at a 12 mm (1/2 inch) application rate for several weeks; then at a lower application rate (approx. 6–8 mm) on a 5-day cycle; then at a 25+ mm (1 inch) rate every three days for two weeks prior to harvest. The intent of adjusting the irrigation rate was to ensure adequate water delivered to the crop. By measuring $ET_a$ directly, the data shown in Figures 7 and 8 clearly show the effect of irrigation timing and persistence of these effects over daily and weekly periods. At site 1, consumptive water use (i.e., ET) only reaches potential ET rates for 3–4 days following irrigation.

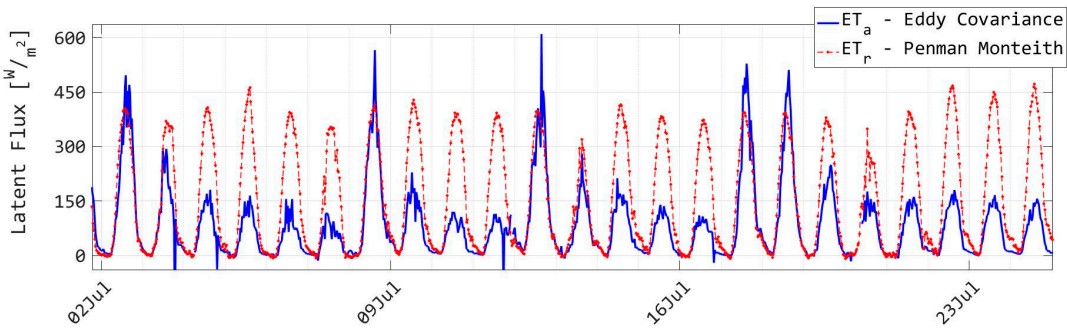

**Figure 7.** Time series of latent flux at Site 1. The variability in ETa followed the 4 to 5-day irrigation cycle (irrigation occurring on 2 July, 8 July, 12 July, 17 July).

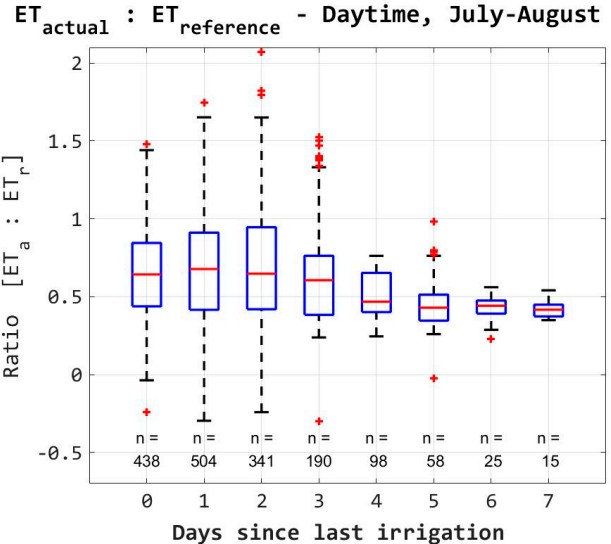

**Figure 8.** Average of ratio of 30-min $ET_a$:$ET_{ref}$, grouped by the number of days since last irrigation (at site 1). Boxes indicate the 1st to 3rd quartile range (25th–75th percentile), with the median value indicated by the red line. Whiskers bound the most extreme values within $\pm\,2.7\sigma$, with outliers outside these bounds indicated by "+" markers.

In many cases, irrigation cannot be applied at optimal levels at all times due to limited water availability, lack of knowledge and time to track crop water demand or simply because of the irrigation system performance. Site 1 has daily reference ET (and calibrated crop coefficients) available for managing the center pivot irrigation but the usefulness of this information is limited because the rate of consumptive use and the irrigation application rate is strongly controlled by soil characteristics.

In this case, measuring $ET_a$ was able to reveal the rapid response in evaporation following irrigation, followed by a 2–3 day return to a lower evaporation rate. Paired with observations of soil moisture and deep percolation and monitoring stress indicators in the crop, this estimate of actual ET provides the last of the terms needed for a water budget used in irrigation scheduling.

### 3.3. Monitoring Cumulative Crop Water Use at Site 2

Relatively longer training times of two weeks were required to reliably estimate $ET_a$ at site 2 (the hazelnut orchard), although as with site 1, a small number of sensor inputs were required. Irrigation was applied on a regular schedule throughout the field trial and no significant variation in crop water stress or weather conditions was observed. Shorter training periods (of one week) resulted in very scattered $ET_{ann}$ values ($R^2 < 0.5$), regardless of the number of sensor inputs that were used. Initially, eight parameters were used: time of day, air temperature and humidity at the canopy top and 3 m above the canopy, soil heat flux, wind speed and net solar radiation (above the canopy). Parameters were added and dropped until a final set of five parameters were selected (time of day, air temperature and humidity at the canopy top, wind speed and net radiation). This input data produced an equally robust network training (Figure 9) to the Penman Monteith equation and larger sets of input variables did not improve on these results.

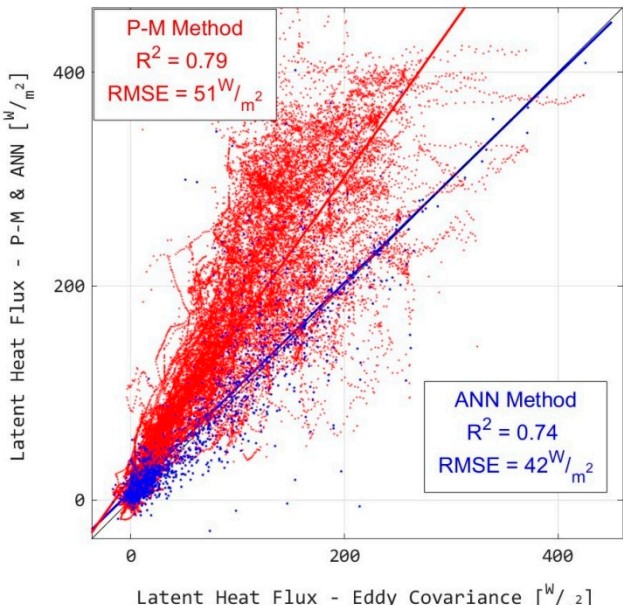

**Figure 9.** Method comparison for ET at site 2 (hazelnut orchard) resulting from two week training period.

The relatively short data collection period of one month may not have been sufficient to evaluate the robustness of the ANN training, although the observed daily and weekly $ET_a$ consistently matched the weekly irrigation rate of 19 mm (0.75 inches), as reported by the irrigation manager (Figure 10). Of particular note in this case is that there are not established hazelnut crop coefficients for use with reference ET. Although the particular solution found here was not validated over an entire growing season, the results demonstrate the advantage of using the ANN method for monitoring actual ET.

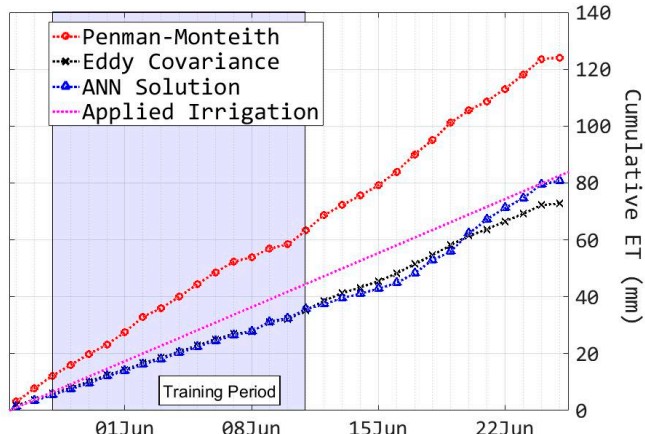

**Figure 10.** Cumulative ET and water use observed at site 2 (hazelnut orchard). The two week training period is shaded in blue.

## 4. Discussion

The ANN method described here facilitates the estimation of $ET_a$ by matching the accessibility of low-cost sensors with the accuracy of direct research methods. By estimating $ET_a$ directly, rather than relying on distant weather data and assuming a uniform crop coefficient, site-specific results of irrigation practices can be evaluated in real time. The measured rate of consumptive water use can then be used to optimize water allocation and irrigation schedules. For crops that do not have established crop coefficients (such as hazelnuts), direct monitoring of actual ET provides an accurate estimate of irrigation demand. Time series of $ET_a$ allow for improved water use efficiency and can also detect crop water stress that may be related to soil water availability, temperature stress and other non-linear crop responses. While causal environmental factors are not necessary revealed by the ANN method, the ability to monitor actual ET directly provides actionable information to guide irrigation allocation.

Because the training process is relatively robust but obscures the underlying physical processes driving ET, implementation of the ANN method requires complementary qualitative and quantitative training supervision. Under applied monitoring conditions, long term co-location of the control measure, by definition, is not available to evaluate the accuracy of the resulting ANN estimates. As a result, numerical training methods must be complemented by observational evaluation that the environmental conditions used for training reflect the overall conditions for the entire monitoring period. Alternately, training must occur over a broad range of conditions. This may imply repeated, shorter (<1 week) training periods that occur during different parts of the growing season (e.g., crop development phases, various weather conditions and different irrigation rates). While it costs relatively little to build a local weather station or to develop an ANN model and validate performance, the greatest cost for valid ANN models will be the trained (human) labor to ensure correct sensor operation and representative field conditions.

The first project goal of optimizing the sensor selection was successful. At both site 1 and site 2, it was possible to produce a robust ANN that accurately reproduced actual ET with just three sensors: a temperature humidity sensor; a radiometer to measure downwelling PAR or shortwave or possibly net radiation; and a wind-speed measurement. The wind speed was not always necessary, although the time of day was also used as an input parameter and does not incur additional cost. With open source microcontrollers (such as Arduino), a complete sensor package could be constructed for as little as 500 USD [37]. Truly low-cost sensors could potentially reduce the cost further, although these were not evaluated in this study. And the ANN method is certainly applicable with typical commercial weather stations that are marketed to farmers.

The second goal to determine the required length of training period requires further study. It is clear that the ANN method presented here is only able to reproduce actual ET for periods that resemble

the training period. As a result, this requires human supervision to match the timing of training to weather, irrigation and field conditions. For established operations or crops with long growing seasons, this would not be a major impediment. On the other hand, the relatively short period of one week, strategically timed, should allow an accurate ANN model to be developed for site specific conditions. Given a total growing season of 20–30 weeks in temperate regions (for a variety of crops), a shared eddy covariance system could service a large number of weather stations each season, potentially making the ~20,000 USD cost for such a system a worthwhile investment for an irrigation district, extension service or crop consultant.

In addition to expanded long term field trials using the ANN method, further work on the application of ANNs to measure ET have been identified. The ANN method developed in Matlab's Neural Network toolbox requires access to expensive and complex software; porting this approach to various open source computational toolboxes/libraries such as existing Python toolboxes [38,39] would allow for much broader access. Using the ANN method with very low-cost sensors such as the LEMS system [37] would also make this method more affordable for in field monitoring. Finally, using the ANN method to utilize agricultural weather networks would facilitate new uses for this valuable, publicly available data, with specific applications such as validation of remote sensing [40], spatial interpolation of reference ET [41] and determination of crop coefficients for new crops.

Efficient irrigation depends on matching application rates to the crop water requirement. Precision irrigation systems also require localized crop water use. Irrigators have many resources to estimate $ET_a$ and crop water use but available weather monitoring data have limited ability to estimate real-time, site-specific ET. Local and real-time ET estimates offer the precision required for science based irrigation to improve allocation decisions and overall efficiency. The ANN method is demonstrably able to convert data from low-cost sensors into site-specific $ET_a$ estimates. Ongoing research will develop ANNs to incorporate other forms of low-cost inputs, such as data from weather networks, satellite imagery and qualitative site observations and manual farm record-keeping. The resulting site-specific algorithms should allow growers to obtain reliable, site-specific $ET_a$ estimates affordably and lead to more efficient and reliable irrigation systems.

**Author Contributions:** Conceptualization, J.K. and E.P.; methodology, J.K.; software, J.K.; validation, J.K.; formal analysis, J.K.; investigation, J.K.; resources, J.K. and E.P.; data curation, J.K.; writing—original draft preparation, J.K.; writing—review and editing, J.K. and E.P.; visualization, J.K.; project administration, J.K.; funding acquisition, J.K. and E.P.

**Funding:** This research was funded by USDA-AFRI, grant number 2017-67012-26125 (http://nifa.usda.gov).

**Acknowledgments:** The authors gratefully acknowledge: Chad Higgins (Oregon State) and Walt Mahaffee (USDA-ARS), who provided support, equipment and ongoing input to the development of ANN and ET monitoring methods; field staff including Taylor Vagher, Willow Walker, Johanna Alexson, Tom DeBell, Payse Smith; and most importantly for the cooperating farms—Greenspring LLC, Jeff Newton and Christensen Farms. Carrie Roever of Northwest Knowledge Network provided assistance with data accessibility. Publication of this article was funded by the University of Idaho Open Access Publishing Fund. The authors acknowledge the contributions made by the three anonymous reviewers, which improved the manuscript significantly.

**Conflicts of Interest:** The authors declare no conflict of interest. The funders had no role in the design of the study; in the collection, analyses or interpretation of data; in the writing of the manuscript or in the decision to publish the results.

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
