# Peer review of "Using Neural Networks to Estimate Site-Specific Crop Evapotranspiration with Low-Cost Sensors"

_agronomy, doi:10.3390/agronomy9020108_

Round 1

Reviewer 1 Report

please see the attached file.

Author Response

Please see the attached document with responses to all of your suggestions. Your review was very thorough and your attention to detail is very much appreciated.

Reviewer 2 Report

In this study, the use of Artificial Neural Networks for the association of the data provided by low cost agrometeorological stations with crop Evapotranspiration obtained by eddy covariance with main target the estimation of crop ET based on low cost agrometeorological stations is investigated. The main idea is that eddy covariance stations could be located near low cost agrometeorological stations for short periods in order to train an ANN that could provide crop ET from the low-cost station for the rest of the period. The idea is very interesting, even if it does not seem to be practical to relocate the eddy covariance stations every little while and to train ANNs for operational purposes. Though, the scientific questions are really interesting and the expected results could provide useful information and be the basis for further study. The applied methodology seems also to be sound and the obtained results very interesting.

On the other hand, the main weaknesses of this study are related to the presentation of the methodology and of the results. I found it really hard to follow the methodology and to distinguish what is tested in each case, what is compared with what, and what were the final results and the final conclusions of each part of the results. While the objectives of the study seem to be clear, I had to read the entire manuscript (methodology, conclusions, results and discussion) to more or less understand how these objectives were met. There were also some additional objectives presented in the methodology or the results sections that makes things a little more confusing.

My opinion I that the manuscript should be carefully revised in order to present more clearly the objectives, the methodology used to achieve them, the obtained results and the final conclusions.

As regards the objectives, some more attention should be given at the last part of the introduction (lines 115-124) where objectives are mixed with results. You may avoid presenting the results in the introduction and try to clearly present the main objectives and any other additional objectives.

Concerning the methodology, the most problematic part seems to be the section 2.2. The first part (lines 221-224) is very confusing and it is not easy to understand what is calculated with what data and for what purpose. Then there is a presentation of the ANN and then again, a presentation of what is calculated.

One possible way to make this easier to understand would be to have a separate section presenting the ANN (e.g. 2.2.). The following section (e.g. 2.3.) could present in detail and clearly what is examined and which is the corresponding methodology for this. As I can understand the main parameters examined are: a) the best set of sensors for low cost stations to estimate ETc and b) the optimum training period and the optimum strategy. However, I also saw some additional objectives in the results (e.g. effect of Irregular Irrigation on Actual ET and some other), if they are important please explain them as well and include them in the general context. It should be clearly presented what is calculated and with what data, the periods that were used for training and for validation, specifically what is compared with what, and the general strategy with specific steps. A graph / table / diagram explaining the methodology could possibly help.

The result section is also confusing similarly to the methodology. It is organized in sections without specific reasoning. For example, the sections titles are: “Optimization of ANN Training”, then “Effect of Irregular Irrigation on Actual ET at Site 1”, “Monitoring Cumulative Crop Water Use at Site 2” and finally “Actual ET from Fully Irrigated Alfalfa”. The sections of the results should be inline with the main scientific questions and / or the main methodological steps. In the current organization the main questions are mixed together and partly answered here and there. The results are also fragmentaly presented and it is very difficult to understand the entire picture. A better organization of the results section is needed (which is actually Results and Discussion).

A table (or tables) summarizing the results (e.g. performance) for each site, each set of sensors, each training duration or any other categorization appropriate could make much easier for the readers to understand what is going on.

The Figures are very good and very informative. Some minor improvements are needed. E.g. in Figures 3, 4, and 8 the slopes and intercepts of the fitted lines should be also presented. It can be indicated, but you should also show the x=y line. Please also clarify if R2 and RMSE are related to the line fit or to the comparison between observed-estimated values. Units should be added in Figs 7 and 12 and please check the legend of Fig. 10.

Finally, in the last section which is mostly conclusions, the key general results of the study should be also clearly and specifically presented along with their importance for practical applications and possible future research. For example, any general results / guidelines concerning the required set of sensors for the “low-cost” stations or the possible strategies that could be followed for the practical application of this research.

Here I would like to mention that it would be interesting to consider the possible cost and the practicality (purchase, installation, maintenance etc.) of the various sets of sensors evaluated. Actually, the setup of the “low-cost” station (lines 156-161) is hardly that of a typical agro-meteorological and a low-cost station. According to the concept it would be interesting to test if a typical agro-meteorological station can provide adequate ETa results with the help of the trained ANN.

Another general comment is that the suggested strategy (lines 482-487) seems not to be very practical (many times relocation of equipment, very high skills required for dealing constantly with equipment, eddy, ANN etc.). This should be discussed.

A final general comment is that the study didn't resulted in a specific low-cost configuration for all purposes as well as a specific training procedure. These should be clarified in order for this study to be useful for practical applications. Though, I would like to state that the obtained results are still very interesting and provide a good base for further study.

Two short specific comments: a) Figure 9, Is the ANN performance better than the simple correlation e.g. a single ratio? b) Lines 346-349, soil moisture spatial variability could be also a problem (see for example Soulis K.X. and Elmaloglou S., 2016, Optimum soil water content sensors placement in drip irrigation scheduling systems: the concept of Time Stable Representative Positions, Journal of Irrigation and Drainage Engineering, ASCE, 10.1061/(ASCE)IR.1943-4774.0001093 , 04016054. or Soulis K.X. and Elmaloglou S., 2018. Optimum soil water content sensors placement for surface drip irrigation scheduling in layered soils. Computers and Electronics in Agriculture, 152, 1-8. doi:10.1016/j.compag.2018.06.052)

Based on the above concerns and comments I believe that the work presented in this study is very good, very laborious, scientifically sound, and generally very interesting. Accordingly, it worth to be published; however, a lot of work (a thorough revision) is still needed in order to make it easier to read and to understand.

Author Response

We appreciate your thorough and constructive review of our manuscript. Please see the attached document in which we address each of your suggestions and how we revised the manuscript in response. In particular, we paid attention to clarifying the methods we employed and the results that were obtained. We also defined the study objectives including the case studies which were included. Due to a miscommunication with a collaborator, we have removed the third case study, but feel that the resulting manuscript is possibly more clear and concise as a result.

Reviewer 3 Report

Dear Authors,

This article examines an interesting research application on water resource management in agriculture, providing a way to estimate ETa directly through the ANN method, by matching the accessibility of low cost sensors with the accuracy of direct research methods (i.e. Eddy Covariance) allowing to evaluate site-specific results of irrigation practices in real time.  The method developed promote an improvement in efficient irrigation by matching application rates to the crop water requirement.

Specific comments are reported in the attached file

Author Response

We appreciate your constructive review of our manuscript. Please see the attached pdf, in which we address each of your suggestions and how we revised the manuscript in response. Our responses can be found in each of the pdf notes. In particular, we paid attention to clarifying the methods we employed and the results that were obtained. We also defined the study objectives including the case studies which were included. Due to a miscommunication with a collaborator, we have removed the third case study, but feel that the resulting manuscript is possibly more clear and concise as a result.

Round 2

Reviewer 2 Report

As I explained in my previous review, in this study, a very interesting idea, namely, the use of Artificial Neural Networks for the association of the data provided by low cost agrometeorological stations with crop Evapotranspiration obtained by eddy covariance with main target the estimation of crop ET based on low cost agrometeorological stations is investigated. The main idea is that eddy covariance stations could be collocated near low cost agrometeorological stations for short periods in order to train an ANN that could provide crop ET from the low-cost station for the rest of the period. This study is very good, very laborious, scientifically sound, and generally very interesting.

Furthermore, the revised version is much easier to read and understand and most of my previous comments were successfully answered / addressed. A problem is that in the answers to my comments there is no specific information on where and how exactly my comments were addressed and there is no manuscript with track changes. For example, in an important comment:

“A table (or tables) summarizing the results (e.g. performance) for each site, each set of sensors, each training duration or any other categorization appropriate could make much easier for the readers to understand what is going on.”

The answer is: “We have added this table.” Though, I cannot locate this table or its description.

Please check what is going on with the table.

Anyway, as I already wrote, the revised manuscript is much better and it is very near to be suitable for publication. I only have very few additional minor comments (consider also the table above) that I have included in the pdf file of the manuscript.

Author Response

Thank you for your additional feedback. I wasn't sure why you couldn't find the revised version with track changes in the previous round (it was included as a .docx file in the zipped folder). For this round, I am attaching one .pdf with my specific responses, and a copy of the revised manuscript with track changes. We appreciate your thorough review and feel that the manuscript has been improved considerably by addressing your suggestions.
